# Evaluation on Egress Safety of Nursing Hospital Considering the Smoke Exhaust System

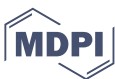

**Seung-Ho Choi [1], Khaliunaa Darkhanbat [2], Inwook Heo [1] and Kang Su Kim [2,\*]**

[1] Department of Architectural Engineering, University of Seoul, 163 Seoulsiripdae-ro, Dongdaemun-gu, Seoul 02504, Korea
[2] Department of Architectural Engineering and the Smart City Interdisciplinary Major Program, University of Seoul, 163 Seoulsiripdae-ro, Dongdaemun-gu, Seoul 02504, Korea
\* Correspondence: kangkim@uos.ac.kr; Tel.: +82-2-6490-2762; Fax: +82-2-6490-2749

**Abstract:** This study aimed to evaluate the egress safety in nursing hospitals based on the capacity of the smoke exhaust system. To this end, the available safe egress time was calculated by analyzing changes in visibility, carbon monoxide, carbon dioxide, oxygen contents, and temperature depending on the fire duration. In addition, an egress simulation was performed using the number of workers (egress guides) and egress delay time as variables, and the required safe egress time was estimated. Based on the results, the egress safety of a prototype nursing hospital was evaluated. In this study, egress safety criteria to evaluate egress safety in a typical nursing hospital were presented, which are expressed in terms of normalized egress guides, the capacity ratio of the smoke exhaust system, and egress delay time. The proposed criteria can be used to evaluate the egress safety of typical nursing hospitals and to prepare complementary measures.

**Keywords:** egress safety criteria; fire dynamic simulator (FDS); nursing hospital; pathfinder; smoke exhaust system





## 1. Introduction

Research has been actively conducted to evaluate the fire safety of buildings via fire and egress simulations. Li et al. [1] performed a fire simulation in a high-rise building using the fire dynamics simulator (FDS) [2] and analyzed changes in visibility and temperature as a function of the height of the smoke exhaust vent. Heo et al. [3] conducted a performance-based egress safety evaluation of studio residential buildings considering the placement of the smoke exhaust system. Ronchi and Nilsson [4] performed an egress simulation of a high-rise building considering human behavior. Maohua et al. [5] and Qin et al. [6] examined fire and egress simulations of metro and subway stations with a large floating population. Hung et al. [7] conducted a study to improve fire safety in small welfare facilities for the elderly. Annunzitata et al. [8] performed fire evacuation drills in a university hospital and conducted an egress simulation reflecting the results. Several other researchers have also performed fire and egress simulations [9–14].

Meanwhile, as the elderly population increases worldwide, the number of nursing hospitals that provide healthcare and welfare services for the elderly has also increased. In the event of a fire, patients in nursing hospitals, who have difficulty moving, have to egress with the help of egress guides. Therefore, because it takes longer for occupants to evacuate when a fire breaks out in a nursing hospital than in other facilities, a relatively large number of casualties can occur [10]. Lee et al. [10] performed fire and egress simulations in nursing hospitals, and Shin et al. [15] conducted a study on measures to secure egress safety in elderly care facilities based on survey results. In addition, studies [16–18] have evaluated egress safety in nursing hospitals. However, in previous research, case studies were mainly conducted for specific nursing hospitals. In such cases, when there were changes in the number of workers (egress guides) and egress delay time, it was cumbersome to perform a

new simulation and re-evaluate the egress safety. Moreover, no analysis has been conducted on the number of egress guides that change depending on the shift patterns and the egress delay time, which vary greatly depending on the building facilities.

In this study, fire simulations were performed to evaluate the egress safety performance of nursing hospitals with smoke exhaust systems. The available safe egress time (ASET) of a prototype nursing hospital was calculated using the capacity of the smoke exhaust system as a variable. In addition, an egress simulation was performed using the number of workers (egress guides) and egress delay time as variables, and the required safe egress time (RSET) was calculated to evaluate the egress safety of the prototype nursing hospital. Based on the results, this study proposed egress safety criteria for a typical nursing hospital with normalized egress guides, capacity ratio of the smoke exhaust system, and egress delay time as variables.

## 2. Simulation Model

### 2.1. Fire Simulation

Figure 1 shows a typical nursing hospital set as a prototype in this study, and Figure 2 shows the FDS [2] model. In this study, fire simulation in the nursing hospital was conducted using the FDS 6.6.0. The target nursing hospital had two floors, a room height of 2.75 m, and a total floor area of 3493.8 m$^2$. The building contained a patient room, worker room, and storage area. In order to derive the egress safety criteria in a conservative way, the fire was assumed to have broken out in the patient's room on the second floor. The two stairways and an elevator were set as egress routes. The length and width of the stairs are 8.0 m and 1.75 m, respectively. Generally, the use of elevators is prohibited in the event of a fire. However, nursing hospitals often have separate elevators that can accommodate wheelchairs or beds for egress during a fire. Therefore, egress using an elevator was considered in this study.

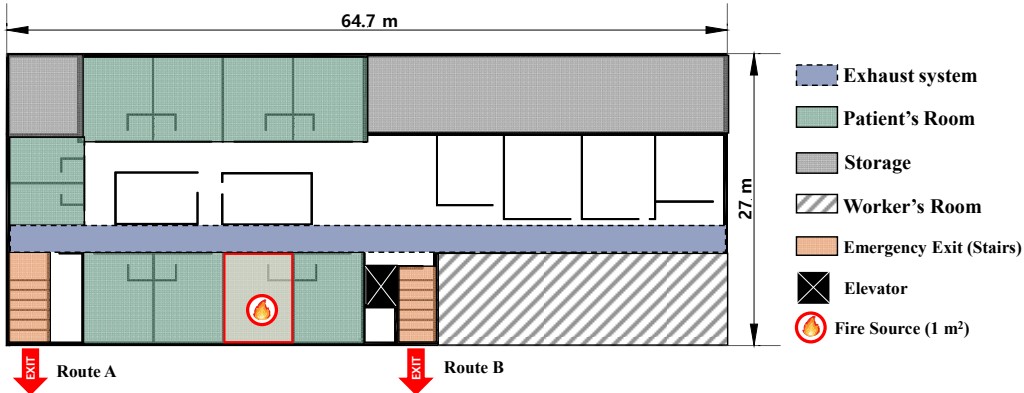

**Figure 1.** Floor plan and location of fire combustion and egress routes.

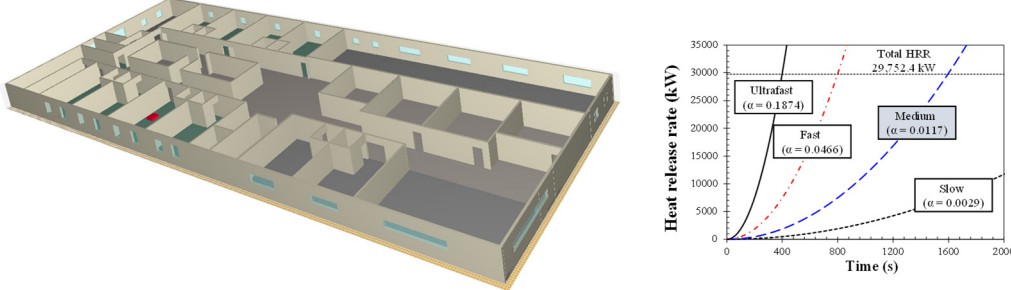

**Figure 2.** Simulation model for fire.

Fire simulation was performed for 800 s, and the initial temperature of the room was assumed to be 20 °C. Before the fire simulation, the mesh sizes [19–21] were carefully checked according to the FDS User Guide Section 6.3.6 Mesh resolution. In more detail, a non-dimensional parameter ($D^*/\delta x$) was set to be between 4 and 16, where $\delta x$ is the nominal size of a mesh cell (m), and $D^*$ is the characteristic fire diameter (m), which can be calculated as follows:

$$D^* = \left( \frac{\dot{Q}}{\rho_\infty c_p T_\infty \sqrt{g}} \right)^{\frac{2}{5}}$$ (1)

where $\dot{Q}$ is the total heat release rate of fire (kW), $\rho_\infty$ is the density of air (1.204 kg/m$^3$), is the specific heat (1.005 kJ/kg·K), $c_p$ is the ambient temperature (293 K), and $T_\infty$ is the acceleration of gravity (9.8 m/s$^2$). For the non-dimensional parameter ($D^*/\delta x$) to have a value of 4 to 16, the mesh size should be set to be between 0.23 m and 0.93 m. In this study, the mesh size was set to be 0.25 m × 0.25 m × 0.27 m, which is the smallest value among those recommended in the FDS User's Guide [2]. Note that analysis for mesh sensitivity check has also been conducted and presented in the Appendix. The beds, chairs, and tables were regarded as combustibles, and the heat release rate (HRR) of each combustible was calculated by referring to the database [22] provided by the National Center for Forensic Science (NCFS). The total HRR was 29,752.4 kW, which was calculated by multiplying the number of combustibles by the maximum value of HRR during the combustion of each combustible. Therefore, the temperature and toxic gases are likely to be overestimated, which gives conservative results on the egress safety. In the T-squared fire method [23], fire modes are classified as slow, medium, fast, and ultrafast based on the fire growth rate. HRR according to the time ($Q$) is defined as

$$Q = \alpha t^2 \text{ (kW)}$$ (2)

where $\alpha$ is the coefficient for fire growth and $t$ is the time. Based on the 'Structural Design for Fire Safety' [23], the fire growth rate of the prototype nursing hospital was set to the medium level, and $\alpha$ was set to 0.0117. The values given in the Society of Fire Protection Engineers handbook [24] were used to determine the properties of the fuel. To present conservative egress criteria, the fuel type was assumed as polyurethane foam (CH1.8). The values of soot yield and CO yield were determined 0.031 and 0.227, respectively, as specified in the Society of Fire Protection Engineers handbook [24]. The burning area in the patient's room was set to 1.0 m × 1.0 m. Measurement devices were installed at a height of 1.8 m for each egress route (Route A, Route B, and Elevator) to measure visibility, CO, O$_2$, and CO$_2$ contents, and temperature.

In this study, the capacity of the smoke exhaust system was classified into seven levels, and the ASETs were compared accordingly. The Korean Fire Protection Association (KFPA) [25] suggests the capacity of a smoke exhaust system considering the floor area of a building. The capacity (S$_c$) of the smoke exhaust system required for the prototype nursing hospital calculated via the KFPA was 12.5 m$^3$/s. Therefore, capacities that were 0%, 25% (3.125 m$^3$/s), 50% (6.25 m$^3$/s), 75% (9.375 m$^3$/s), 100% (12.5 m$^3$/s), 125% (15.625 m$^3$/s), and 150% (18.75 m$^3$/s) of the prototype capacity were considered in the simulation. All smoke exhaust systems were assumed to be installed in the hallway. In case of a fire, most doors and windows are burned or destroyed, which is in fact the worst condition. Therefore, in this study, the doors and windows of all compartments were assumed to be open, which gives conservative results.

### 2.2. Egress Simulation

In this study, an egress simulation was performed using Pathfinder [26]. The time required for all occupants in the egressed building was calculated as the RSET. Figure 3 shows the Pathfinder model for the egress simulation. The workers (egress guides) who were on the 1st floor can possibly go up to the 2nd floor and help for egress of the patients.

In order to consider such egress characteristics, the egress simulation was performed for two floors. The Enforcement Rule of Welfare of the Senior Citizens Act [27] stipulates that a hospital building should have a minimum area of 23.6 m$^2$ per patient. Therefore, considering the area of the prototype building, the number of patients was assumed to be 150. Table 1 shows the proportion of patients according to severity and number of patients by egress type. The following assumptions were drawn. In the event of a fire, the proportion of patients with 'medical highest' severity was 1.8% (3 patients), and they use beds to egress with the help of an egress guide because of their inability to move independently. The proportion of patients with 'medical high' severity was 22.9% (35 patients), of which thirteen use beds to egress with the help of an egress guide, and the remaining 22 use wheelchairs to egress with the help of an egress guide. Additionally, patients with 'medical middle' severity use wheelchairs to egress with the help of an egress guide, whereas patients with 'medical low' severity use wheelchairs to egress independently. The proportion of patients with 'Etc. level' severity was 27.7% (41 patients), and they could evacuate on foot.

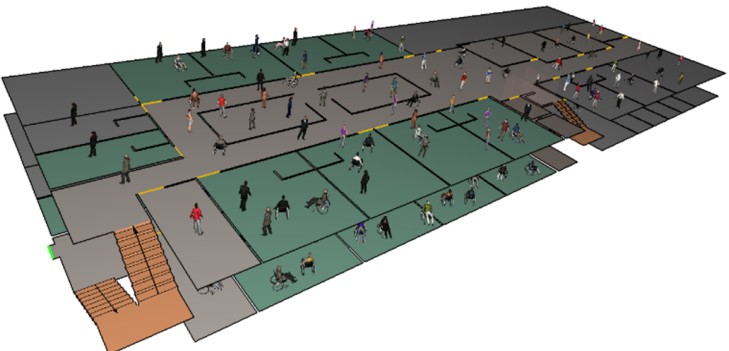

**Figure 3.** Simulation model for egress.

**Table 1.** Number of patients according to severity and egress type.

| Status | Ratio (%) | Number of Patients | Egress Type |
|---|---|---|---|
| Medical highest | 1.8 | 3 | Bed (aided-egress) |
| Medical high | 22.9 | 13 | Bed (aided-egress) |
|  |  | 22 | Wheelchair (aided-egress) |
| Medical middle | 24.8 | 37 | Wheelchair (aided-egress) |
| Medical low | 22.8 | 34 | Wheelchair (self-egress) |
| Etc. | 27.7 | 41 | Walk (self-egress) |
| Total | 100 | 150 |  |

Table 2 shows the width, velocity, and preparation time of workers (egress guides) and patients in the nursing hospital. The corresponding values were determined based on the results of a study conducted by Lee et al. [10].

**Table 2.** Characteristics of workers and patients.

| | Category | Width (cm) | Velocity (m/s) | Preparation Time (s) |
|---|---|---|---|---|
| Workers | Care worker | 359 | 1.3 | - |
| | Doctor | 403 | 1.5 | |
| | Nurse | 356 | | |
| | Physical therapist/social worker | 403 | | |
| Patients | Walk (self-egress) | 500 | 0.5 | |
| | Wheelchair (self-egress) | 700 | 0.8 | 15 |
| | Wheelchair (aided-egress) | | 1.5 | |
| | Bed (aided egress) | 720 | 0.6 | 25 |

Table 3 presents the number of workers (egress guides). In Pathfinder, egress simulation can perform by specifying the characteristics of occupants through the "behaviors" option. Among the "behavior" options, the "assistant model" is a characteristic that helps others' egress. In this study, the workers (egress guides) were modeled as the "assistant model" so that they can help the patients' egress. The Enforcement Rule of Welfare of Senior Citizens Act [27] specifies the number of care workers, doctors, physical therapists, social workers, and nurses required according to the number of patients. However, in nursing hospitals where employees work in shifts, the actual number of occupants varies depending on the shift, and the number of night-shift workers may be even smaller. Therefore, in this study, an egress simulation was performed using the number of workers on duty as a variable. The minimum number of workers (egress guides) on duty was set to ten, and the analysis was performed by increasing the number of workers by five each time. The maximum number of workers considered was 75, which was based on the values given in the Enforcement Rule of Welfare of Senior Citizens Act [27], assuming 150 patients.

**Table 3.** Number of workers (egress guides).

| Number of Egress Guides | Care Worker | Doctor | Physical Therapist | Social Worker | Nurse | 1st Floor Occupants | 2nd Floor Occupants | Total Occupants |
|---|---|---|---|---|---|---|---|---|
| 10 | 5 | 1 | 1 | 1 | 2 | 69 | 91 | 160 |
| 15 | 10 | | | | | 74 | 91 | 165 |
| 20 | 15 | | | | | 76 | 94 | 170 |
| 25 | 20 | | | | | 79 | 96 | 175 |
| 30 | 20 | 2 | 2 | 2 | 4 | 82 | 98 | 180 |
| 35 | 20 | 3 | 3 | | 7 | 85 | 100 | 185 |
| 40 | 25 | | | | | 87 | 103 | 190 |
| 45 | 30 | | | | | 89 | 106 | 195 |
| 50 | 35 | | | | | 91 | 109 | 200 |
| 55 | 40 | | | | | 93 | 112 | 205 |
| 60 | 45 | | | | | 95 | 115 | 210 |
| 65 | 50 | | | | | 97 | 118 | 215 |
| 70 | 55 | | | | | 99 | 121 | 220 |
| 75 | 60 | | | | | 101 | 124 | 225 |

The British Standard Institute [28] and Ministry of Public Safety and Security (MPSS) [29] suggest an egress delay time for each building. The egress delay time refers to the time required for occupants to start egress after a fire breaks out. This includes the time it takes to detect the fire, raise an alarm, and move after comprehending the situation. In this study, simulations were performed by setting the egress delay time from 0 to 300 s at intervals of 60 s, and the egress safety was evaluated accordingly.

## 3. Simulation Results

### 3.1. Fire Simulation Results

Figure 4 shows the variation in smoke behavior according to the capacity of the smoke exhaust system at fire durations of 200, 400, 600, and 800 s. Without the smoke exhaust system, the room where the fire broke out was filled with smoke, which spread to most of the hallway approximately 200 s after it began. After 400 s, smoke spread to all rooms on the floor. When the capacity of the smoke exhaust system was 6.25 $m^3$/s, the amount of smoke was negligible in the hallway 200 s after the fire; however, the hallway was full of smoke 400 s after the fire. Overall, the spread of smoke decreased significantly as the capacity of the smoke exhaust system increased.

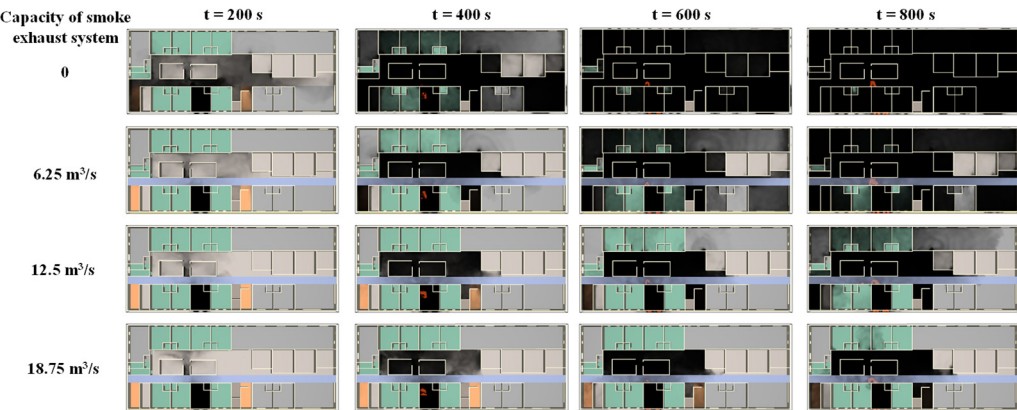

**Figure 4.** Smoke behavior at 2nd floor according to capacity of smoke exhaust system.

The National Fire Protection Association (NFPA) [30] outlines the main factors that directly affect occupant safety in the event of a fire and their tenability criteria. In this study, visibility, CO, $CO_2$, and $O_2$ contents, and temperature at the egress routes were measured at specific durations from the FDS results, and the time by which each factor exceeded the tenability criteria was calculated. Finally, the lowest value among the times calculated via the five factors was determined as the ASET of the prototype nursing hospital. Figure 5 shows the results of visibility, CO, $CO_2$, and $O_2$ contents, and temperature at each egress route (Route A, Route B, and elevator) in the case without a smoke exhaust system. The graph also shows the tenability criteria specified in NFPA [30]. As shown in Figure 5a, when there was no smoke exhaust system, the visibility according to the fire duration time (*t*) reached the tenability criterion (5 m) in the following order: Route A (212 s), Route B (263 s) and elevator (273 s). As shown in Figure 5b, the CO content reached the tenability criterion (1400 ppm) in the following order: Route A (460 s), Route B (611 s), and elevator (670 s). Figure 5c,e show that Route A reached the tenability criteria for $CO_2$ content and temperature (5%, 60 °C) at 700 and 750 s, respectively, and Route B and the elevator did not reach the tenability criteria within 800 s of analysis. As shown in Figure 5d, $O_2$ content did not reach the tenability criteria for all egress routes within the analysis period. In the absence of a smoke exhaust system, the ASET for occupants was predominantly determined by a decrease in visibility, and the ASET was calculated at 212 s.

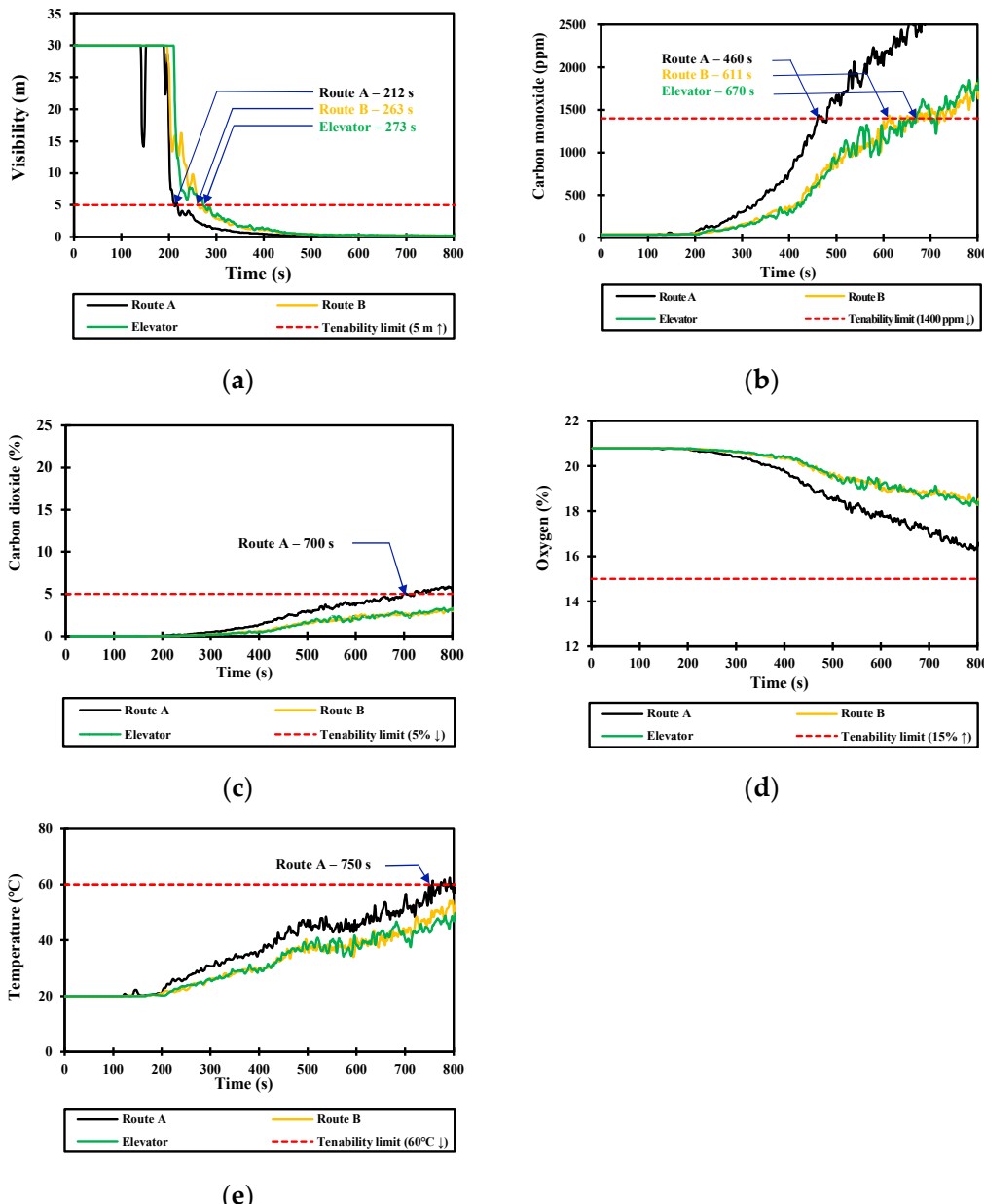

**Figure 5.** (**a**) Visibility; (**b**) carbon monoxide; (**c**) carbon dioxide; (**d**) oxygen; (**e**) temperature.

Figure 6 shows the visibility results according to the capacity of the smoke exhaust system at each egress route. When the capacity of the smoke exhaust system was 3.125 m³/s, the tenability criterion (5 m) was first reached in Route A (297 s), and the tenability criteria were reached at 345 s for both Route B and the elevator. When the capacity of the smoke exhaust system was 3.125 m³/s, the ASET was determined to be 297 s, indicating an increase of 85 s compared with the case without a smoke exhaust system. When the capacity of the smoke exhaust system was 6.25 m³/s, the visibility criteria were reached first in Route B, and when it was 9.375 m³/s or more, the visibility criteria were reached first in the elevator. As the capacity of the smoke exhaust system increased, the ASET also tended to increase.

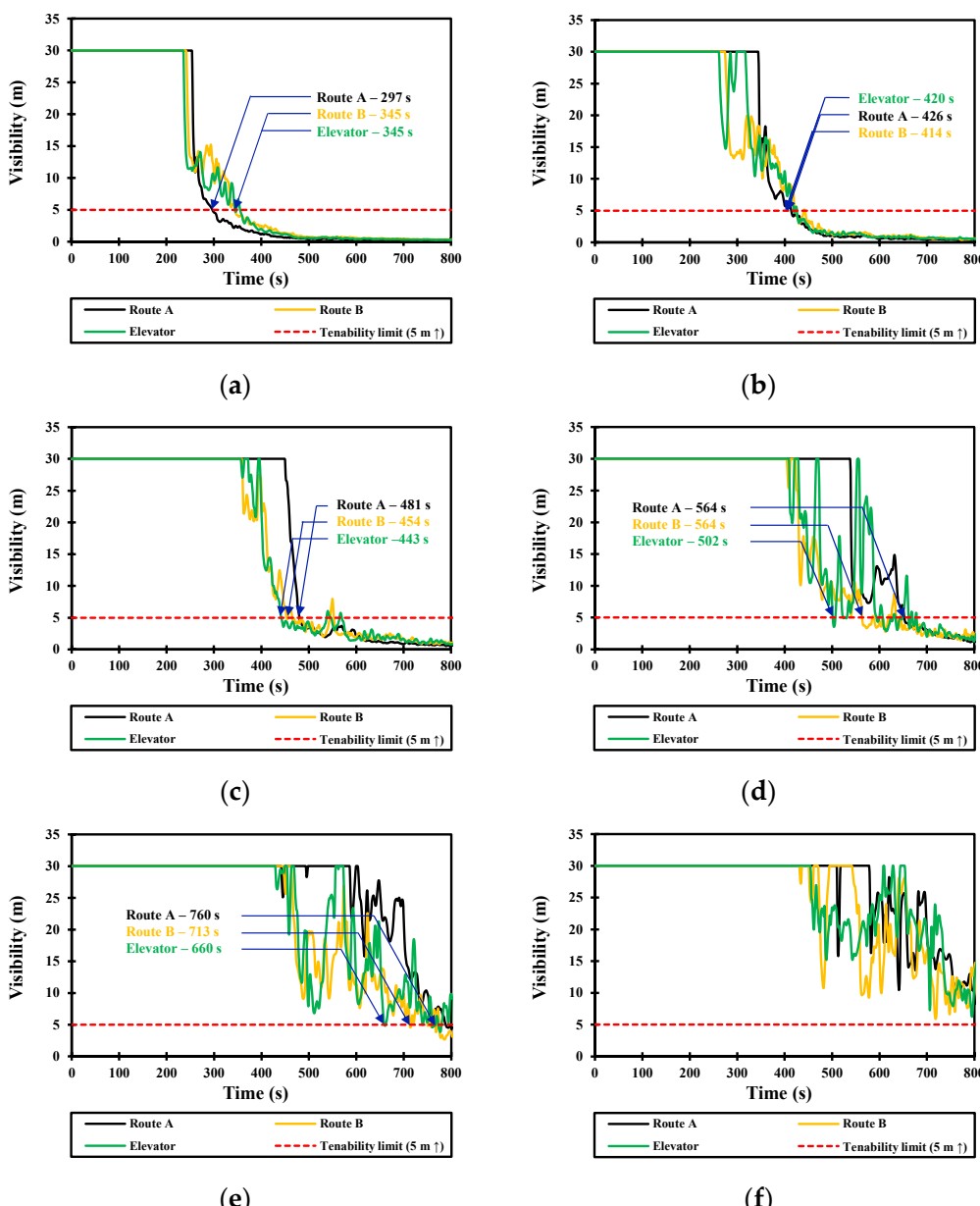

**Figure 6.** (**a**) Capacity of smoke exhaust system 3.125 m³/s; (**b**) capacity of smoke exhaust system 6.25 m³/s; (**c**) capacity of smoke exhaust system 9.375 m³/s; (**d**) capacity of smoke exhaust system 12.5 m³/s; (**e**) capacity of smoke exhaust system 15.625 m³/s; (**f**) capacity of smoke exhaust system 18.75 m³/s.

Table 4 summarizes the ASET results according to the capacity of the smoke exhaust system. When the capacity of the smoke exhaust system was 6.25 m³/s or higher, factors other than visibility did not reach the tenability criteria within the analysis period.

**Table 4.** Available safe egress time (ASET).

| Capacity of Smoke Exhaust System | 0 (m³/s) | 3.125 (m³/s) | 6.25 (m³/s) | 9.375 (m³/s) | 12.5 (m³/s) | 15.625 (m³/s) | 18.75 (m³/s) |
|---|---|---|---|---|---|---|---|
| Visibility | 212 | 297 | 414 | 443 | 502 | 660 | - |
| Carbon monoxide | 460 | 708 | - | - | - | - | - |

**Table 4.** *Cont.*

| Capacity of Smoke Exhaust System | 0 (m³/s) | 3.125 (m³/s) | 6.25 (m³/s) | 9.375 (m³/s) | 12.5 (m³/s) | 15.625 (m³/s) | 18.75 (m³/s) |
|---|---|---|---|---|---|---|---|
| Carbon dioxide | 700 | - | - | - | - | - | - |
| Oxygen | - | - | - | - | - | - | - |
| Temperature | 750 | 800 | - | - | - | - | - |

*3.2. Egress Simulation Results*

Figure 7 shows the number of survivors according to the fire duration and egress delay times for cases in which the total number of evacuees was 190. There were 150 patients, 40 egress guides, and 103 evacuees in the study. The time when all occupants completed the egress was calculated as the RSET for each case. In a situation where the number of evacuees was constant, the egress pattern was similar. Therefore, all graphs showed similar trends, and RSET tended to be offset according to the egress delay time.

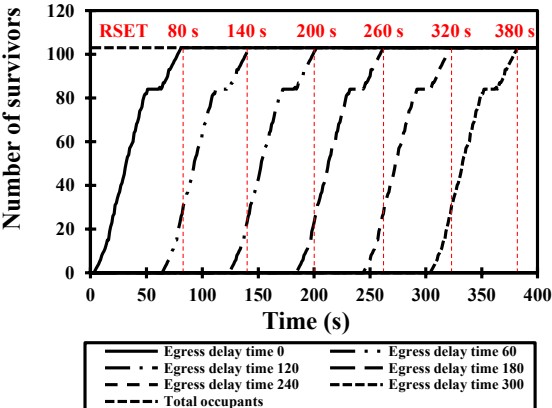

**Figure 7.** Simulation results (total evacuees = 103).

Table 5 shows the RSET of the prototype building equipped without a smoke exhaust system according to the egress delay time and number of egress guides. When the number of egress guides increased, the RSET decreased. When the egress delay time increased, the RSET increased. However, if the number of egress guides increased while the number of patients was fixed, the total number of evacuees increased. Therefore, in some cases, although there were more egress guides, RSET increased owing to the increase in the total number of evacuees.

**Table 5.** Required safe egress time (RSET).

| Number of Egress Guides | Egress Delay Time (s) | | | | | |
|---|---|---|---|---|---|---|
| | 0 | 60 | 120 | 180 | 240 | 300 |
| 10 | 456 | 516 | 576 | 636 | 696 | 756 |
| 15 | 331 | 391 | 451 | 511 | 571 | 631 |
| 20 | 166 | 226 | 286 | 346 | 406 | 466 |
| 25 | 165 | 225 | 285 | 345 | 405 | 465 |
| 30 | 162 | 222 | 282 | 342 | 402 | 462 |
| 35 | 67 | 127 | 187 | 247 | 307 | 367 |
| 40 | 80 | 140 | 200 | 260 | 320 | 380 |
| 45 | 70 | 130 | 190 | 250 | 310 | 370 |

**Table 5.** *Cont.*

| Number of Egress Guides | Egress Delay Time (s) | | | | | |
|---|---|---|---|---|---|---|
| | **0** | **60** | **120** | **180** | **240** | **300** |
| 50 | 71 | 131 | 191 | 251 | 311 | 371 |
| 55 | 68 | 128 | 188 | 248 | 308 | 368 |
| 60 | 63 | 123 | 183 | 243 | 303 | 363 |
| 65 | 59 | 119 | 179 | 239 | 299 | 359 |
| 70 | 69 | 129 | 189 | 249 | 309 | 369 |
| 75 | 61 | 121 | 181 | 241 | 301 | 361 |

## 4. Egress Safety Evaluation

Table 6 shows the egress safety evaluation results for the nursing hospital without the smoke exhaust system. The number of people who failed to evacuate when the fire duration reached ASET was presented according to the number of egress guides and delay time. When there were ten egress guides, even if they started to egress immediately after the fire, four people failed to evacuate. When there were 20, all occupants successfully egressed only when the egress delay time was 0 s, whereas two people failed to egress when the egress delay time was 60 s. Therefore, at least 20 workers must be on duty even during the night shift to successfully egress during a fire. When the egress delay time was the same, the number of people who failed to egress decreased as the number of egress guides increased. However, when the number of egress guides increased while the number of patients was fixed, the total number of evacuees increased. Therefore, in some cases, although there were more egress guides, the number of casualties increased. It was found that because the ASET was 212 s, no one could successfully egress when the egress delay time exceeded 240 s. Meanwhile, as the number of egress guides increases, it is expected that the egress delay time, including the time it takes to detect a fire and raise an alarm, will decrease, resulting in increased egress safety.

**Table 6.** Number of casualties for nursing hospital without smoke exhaust system.

| Number of Egress Guides | Egress Delay Time (s) | | | | | |
|---|---|---|---|---|---|---|
| | **0** | **60** | **120** | **180** | **240** | **300** |
| 10 | 4 | 6 | 5 | 48 | 91 | 91 |
| 15 | 3 | 8 | 10 | 44 | 91 | 91 |
| 20 | 0 | 2 | 8 | 44 | 94 | 94 |
| 25 | 0 | 2 | 8 | 46 | 96 | 96 |
| 30 | 0 | 2 | 7 | 55 | 98 | 98 |
| 35 | 0 | 0 | 0 | 56 | 100 | 100 |
| 40 | 0 | 0 | 0 | 52 | 103 | 103 |
| 45 | 0 | 0 | 0 | 58 | 106 | 106 |
| 50 | 0 | 0 | 0 | 64 | 109 | 109 |
| 55 | 0 | 0 | 0 | 60 | 112 | 112 |
| 60 | 0 | 0 | 0 | 54 | 115 | 115 |
| 65 | 0 | 0 | 0 | 51 | 118 | 118 |
| 70 | 0 | 0 | 0 | 55 | 121 | 121 |
| 75 | 0 | 0 | 0 | 70 | 124 | 124 |

Figure 8 shows the success or failure of egress and the number of casualties for each case. In the graph, the *x*-axis represents the capacity of the smoke exhaust system, whereas the *y*-axis represents the number of egress guides. If all occupants egressed successfully, it was determined to be "safe." On the other hand, if even one person failed to egress, it was determined to be "unsafe," and the number of casualties was listed in this case. Meanwhile, as the capacity of the smoke exhaust system increased, the ASET increased, and the number of occupants who failed to egress decreased. It was also found that when the capacity of the smoke exhaust system was 15.625 m$^3$/s, although the egress delay time was 300 s, there would be no casualties if at least 15 workers were on duty. However, when the number of workers (egress guides) was ten, the egress delay time should be less than 180 s to prevent casualties. However, when the capacity of the smoke exhaust system was 18.75 m$^3$/s, no casualties occurred even if the egress delay time was 300 s, regardless of the number of workers.

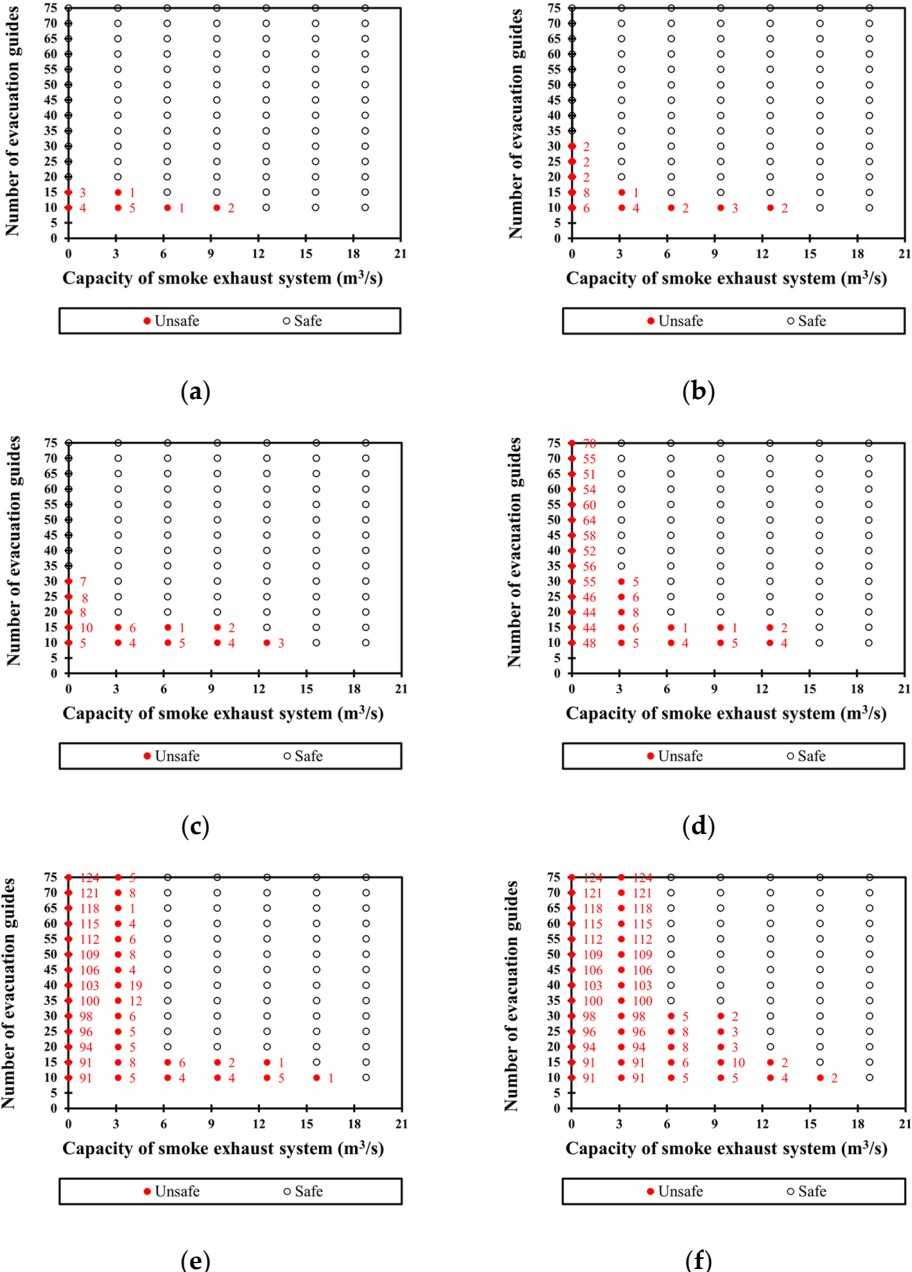

**Figure 8.** (**a**) Egress delay time 0; (**b**) egress delay time 60; (**c**) egress delay time 120; (**d**) egress delay time 180; (**e**) egress delay time 240; (**f**) egress delay time 300.

## 5. Egress Safety Criteria

Figure 9 shows the egress safety criteria proposed in this study. In the graph, the *x*-axis represents the capacity ratio of the smoke exhaust system, whereas the y-axis represents the normalized number of egress guides. The capacity ratio of the smoke exhaust system (1/hour) was calculated by dividing the capacity of the smoke exhaust system by the volume of one floor in the prototype building. The normalized number of egress guides was calculated by dividing the number of egress guides by the total number of patients. Generally, the number of occupants in a nursing hospital can be determined according to the building size, and the number of employees and workers on duty can be determined by considering the number of patients. Therefore, the normalized number of egress guides calculated by reflecting the building size and number of patients is an objective value and can be used to determine the egress safety criteria. Furthermore, the capacity of the smoke exhaust system is represented in the form of volume per unit time, and can be quantified by dividing it by the volume of one floor in the building. Therefore, the capacity ratio of the smoke exhaust system can be used to determine the egress safety criteria as an objective value reflecting the size of the building and smoke exhaust system. The line shown in Figure 9 represents the egress safety criteria used to determine the success or failure of egress according to the capacity of the smoke exhaust system and the number of egress guides in a typical nursing hospital. Data points below this line (shaded area) denote egress failure, whereas those above denote success. The simulation results showed that the egress delay time significantly influences egress safety. This study presents egress safety criteria with a tendency for the unsafe zone to increase with an increase in egress delay time, as shown in Figure 9. The proposed criteria can be used to confirm egress safety in a typical nursing hospital and help in devising complementary measures. If a nursing hospital is judged to be unsafe according to these criteria, the number of egress guides on duty or the capacity of the smoke exhaust system can be increased. Moreover, broadcasting equipment or CCTV can assist in enhancing workers' proficiency levels in handling fire situations, thereby reducing egress delay time. Egress safety can be improved by incorporating a combination of the three methods outlined above. In the case of an actual nursing hospital, a method can be selected in terms of cost and efficiency to increase egress safety.

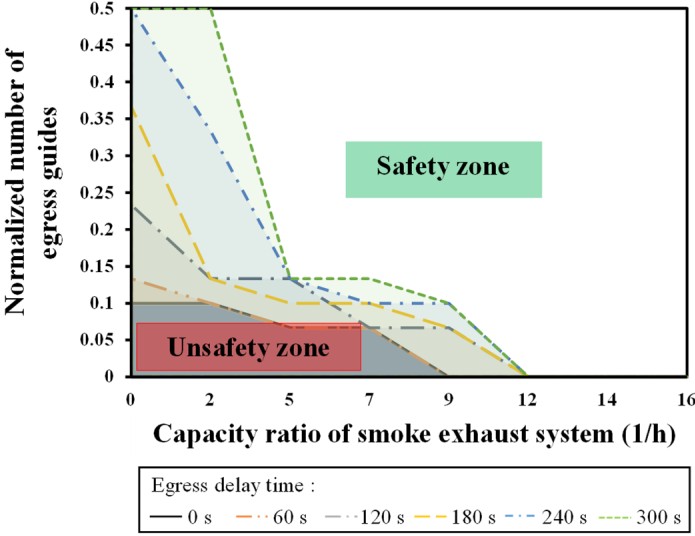

**Figure 9.** Egress safety criteria for nursing hospitals considering smoke exhaust system.

## 6. Summary and Conclusions

This study evaluated the egress safety of a prototype nursing hospital according to the number of egress guides, capacity of the smoke exhaust system, and egress delay time. The following conclusions were drawn:

1. The changes in the main factors (visibility, CO, $CO_2$, and $O_2$ contents, and temperature) according to the fire duration time ($t$) of a prototype nursing hospital were analyzed using the capacity of the smoke exhaust system as a variable. The analysis revealed that as the capacity of the smoke exhaust system increased, the time by which each factor exceeded the tenability criteria increased, and the ASET also increased. The most important factor determining ASET was the decrease in visibility due to the spread of smoke.
2. The egress simulation of a prototype nursing hospital was performed considering occupant characteristics. The simulation revealed that the RSET tended to increase as the number of egress guides decreased and the egress delay time increased.
3. The egress safety of the nursing hospital was evaluated according to the number of egress guides, capacity of the smoke exhaust system, and egress delay time. The minimum number of workers on duty required for each case was calculated. The results confirmed that if there is no smoke exhaust system, at least 20 egress guides are needed; however, when an exhaust system was installed, the number of egress guides could be reduced depending on its capacity.
4. This study proposed egress safety criteria for a typical nursing hospital with the normalized number of egress guides, capacity ratio of the smoke exhaust system, and egress delay time as variables. The proposed criteria can be used to evaluate the egress safety performance of a typical nursing hospital without having to perform a new simulation. In the case of an actual nursing hospital, a method can be selected in terms of cost and efficiency to increase egress safety.

**Author Contributions:** Writing—original draft, S.-H.C.; investigation, K.D.; investigation, I.H.; writing—review and editing, K.S.K. All authors have read and agreed to the published version of the manuscript.

**Funding:** This work is supported by the Korea Agency for Infrastructure Technology Advancement (KAIA) grant funded by the Ministry of Land, Infrastructure, and Transport (Grant 22RMPP-C163162-02).

**Institutional Review Board Statement:** Not applicable.

**Informed Consent Statement:** Not applicable.

**Data Availability Statement:** The data presented in this study are available on request to the corresponding author.

**Conflicts of Interest:** The authors declare no conflict of interest.

## Appendix A. Analysis Results for Mesh Sensitivity Check

The fire simulation has been performed with the mesh size of 0.20 m × 0.20 m × 0.20 m, 0.25 m × 0.25 m × 0.27 m, 0.30 m × 0.30 m × 0.30 m, 0.40 m × 0.40 m × 0.41 m, 0.50 m × 0.50 m × 0.46 m, whose results are shown in Figures A1 and A2. Note that the mesh size smaller than 0.20 m × 0.20 m × 0.20 m is very difficult due to the extremely high computational loads. The simulation results showed no significant difference in ASET or any strong bias according to the mesh size. Therefore, the mesh size was set to be 0.25 m × 0.25 m × 0.27 m.

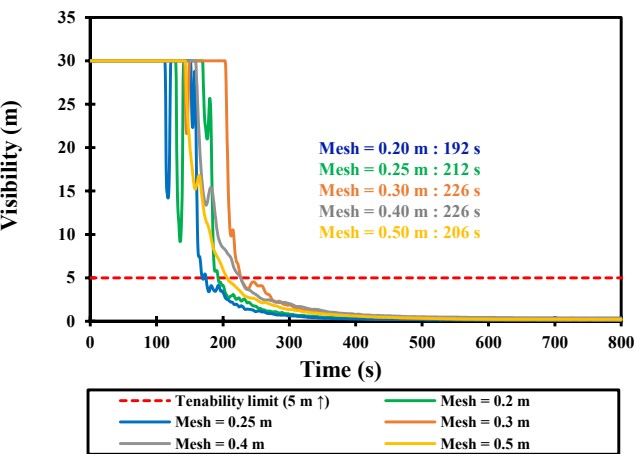

**Figure A1.** Visibility results according to the mesh size.

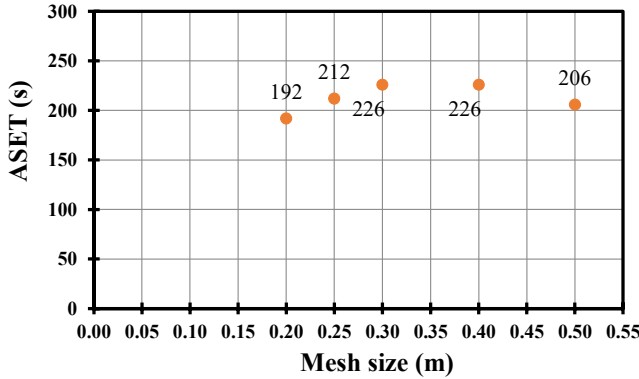

**Figure A2.** ASET according to the mesh size.

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
