# Peer review of "Evaluation on Egress Safety of Nursing Hospital Considering the Smoke Exhaust System"

_fire, doi:10.3390/fire5040120_

Round 1

Reviewer 1 Report

This paper evaluated the egress safety of a nursing hospital by using numerical simulation methods with FDS and Pathfinder. In general, this is already a well-studied topic, other than the building type of nursing hospital itself. This is also the major problem that I am concerned about. Such researches, to be honest, are more like a performance-based design report rather than a scientific paper. Because all the results are strongly related to their own set-ups, i.e., the building type, fire HRRs, fire prevention facility, etc. No new insight into general results can be obtained. I would suggest a major revision and let the editor to make the final decision.

Some other comments.

1) The visibility, CO, O2, and CO2 concentration are strongly related to the CO and soot yields. The reason of choosing these values should be carefully given.

2) A time-depended HRR evolution trend should be given in order to better understand the setups.

3) A mesh sensitivity analysis should be added to illustrate the rationale of choosing the current mesh resolution.

4) The burning area in the patient's room was set to 1.0 m × 1.0 m. And the total HRR was 29,752.4 kW. Is it reasonable to set such a large fire on such a small burning area? Are there any difference for the simulation results if you set a larger area?

5) What is the location of the fire? Are there any difference for the simulation results if you choose different locations?

6) Why the simulation model for fire only shows a single floor but the simulation model for egress has two floors?

7) The models for the simulation of workers and patients should be explained. Could Pathfinder simulate the process of workers helping the evacuation of patients, and how? Otherwise, the increasing of the number of workers is meaningless.

Reviewer 2 Report

In this study, fire simulation was performed to evaluate the evacuation safety performance of a nursing hospital equipped with a smoke exhaust device. The ASET of the nursing hospital was calculated using the capacity of the smoke exhaust device as a variable, and the RSET was calculated to evaluate the safety of evacuation. In general, the goal of PBD is to identify and improve the hazards of the designed structure. Therefore, in order for PBD to become a research article, standardization of standards for target buildings such as subways and tunnels must precede. This study contains serious errors in originality and ripples that a research thesis should have as a general report to improve the level of review and safety in the field of fire safety in general buildings.

Reviewer 3 Report

This study evaluated the egress safety of a prototype nursing hospital according to the number of egress guides, capacity of the smoke exhaust system, and egress delay time. The suggestions are as follows:

1. How does the study determine the grid size? Has the grid independence analysis been conducted?

2. Is it reasonable that the doors and windows of all compartments are assumed to be open in this study?

3. Does the setting of the number of patients and workers in the model match the real situation?

4. The English writing of this paper needs to be checked carefully.

Reviewer 4 Report

Review in pdf.

Round 2

Reviewer 1 Report

The authors have addressed most of my comments. However, the paper still got the following issue to be resolved.

A simple D* formula is not enough, a complete section mesh resolution discussion (mesh sensitivity analysis) should be added in my opinion. This is very important for purely numerical simulation work. Without that, invalidated results doesn’t make sense. See these papers for instance.

https://doi.org/10.1080/00102202.2016.1139367 https://doi.org/10.1016/j.ijthermalsci.2019.03.016 https://doi.org/10.1016/j.energy.2020.119470

Still the exact location of the fire chosen and the reason should be added to the paper. I am still not convinced to be honest, if the reason is to ‘derive the egress safety criteria in a conservative way’, why the authors why not put the fire at the first floor then?

What exactly the egress safety criteria is should be summarized and put forward clearly both in the Abstract and Conclusion parts, rather than the ambiguous words. 

Reviewer 4 Report

The response of the authors to the comments and suggestions are all in satisfactory. I suggest that those responses from comments and suggestions maybe included in the manuscript. The scientific merit of this study is still very limited, but it provides practical data and therefore I recommend publication.
